# Management of High-Risk Hypercholesterolemic Patients and PCSK9 Inhibitors Reimbursement Policies: Data from a Cohort of Italian Hypercholesterolemic Outpatients

**DOI:** 10.3390/jcm11164701

**Published:** 2022-08-11

**Authors:** Federica Fogacci, Marina Giovannini, Elisa Grandi, Egidio Imbalzano, Daniela Degli Esposti, Claudio Borghi, Arrigo F. G. Cicero

**Affiliations:** 1Hypertension and Atherosclerosis Research Group, Medical and Surgical Sciences Department, Alma Mater Studiorum University of Bologna, 40138 Bologna, Italy; 2Cardiovascular Medicine Unit, IRCCS Azienda Ospedaliero-Universitaria di Bologna, 40138 Bologna, Italy; 3Department of Clinical and Experimental Medicine, University of Messina, 98122 Messina, Italy

**Keywords:** cardiovascular risk, tailored medicine, hypercholesterolemia, LDL-C, PCSK9 inhibitors, LDL-C goal

## Abstract

Proprotein convertase subtilisin/kexin type 9 (PCSK9) inhibitors are effective and safe lipid-lowering treatments (LLT). The primary endpoint of the study was to assess the prevalence of patients eligible for treatment with PCSK9 inhibitors in a real-life clinical setting in Italy before and after the recent enlargement of reimbursement criteria. For this study, we consecutively considered the clinical record forms of 6231 outpatients consecutively admitted at the Lipid Clinic of the University Hospital of Bologna (Italy). Patients were stratified according to whether they were allowed or not allowed to access to treatment with PCSK9 inhibitors based on national prescription criteria and reimbursement rules issued by the Italian Medicines Agency (AIFA). According to the indications of the European Medicines Agency (EMA), 986 patients were candidates to treatment with PCSK9 inhibitors. However, following the prescription criteria issued by AIFA, only 180 patients were allowed to access to PCSK9 inhibitors before reimbursement criteria enlargement while 322 (+14.4%) with the current ones. Based on our observations, low-cost tailored therapeutic interventions for individual patients can significantly reduce the number of patients potentially needing treatment with PCSK9 inhibitors among those who are not allowed to access to the treatment. The application of enlarged reimbursement criteria for PCSK9 inhibitors could mildly improve possibility to adequately manage high-risk hypercholesterolemic subjects in the setting of an outpatient lipid clinic.

## 1. Introduction

Among the most recently developed drugs able to significantly improve serum levels of low-density lipoprotein cholesterol (LDL-C), there are specific human monoclonal antibodies inhibiting the proprotein convertase subtilisin/kexin type 9 (PCSK9) and preventing the LDL receptor degradation in the liver [1,2]. Their efficacy and safety has been largely demonstrated in a number of clinical trials, and, in particular, in two multicenter international outcome studies: the Further Cardiovascular Outcomes Research With PCSK9 Inhibition in Subjects With Elevated Risk (FOURIER) and the Evaluation of Cardiovascular Outcomes After an Acute Coronary Syndrome During Treatment With Alirocumab (ODISSEY OUTCOMES) [3,4].

The European Medicine Agency (EMA) approved the use of PCSK9 inhibitors in patients affected by familial hypercholesterolemia (FH), non-familial hypercholesterolemia, or mixed dyslipidaemias as add-on therapy in patients unable to reach the LDL-C target despite high-intensity lipid-lowering treatment (LLT) (or non-statin LLT in case of statin intolerance). Moreover, PCSK9 inhibitors are indicated and provide new options for the treatment of hypercholesterolemia in adults with established atherosclerotic CVD (i.e., myocardial infarction, stroke, or peripheral arterial disease) to reduce CV risk by lowering LDL-C levels, as an adjunct to the maximum tolerated dose of a statin with or without other LLT or, alone or in combination with other LLT in statin-intolerant patients. Finally, the PCSK9 inhibitor evolocumab can also be used in patients > 12 years affected by homozygous FH (HoFH) [5].

The US Food and Drug Administration (FDA) also approved the use of PCSK9 inhibitors in similar clinical settings [6]. However, because of the relatively high cost of these drugs, national governments and insurances have adopted different policies regarding reimbursement and prescription authorization [7,8].

The primary endpoint of the study was to assess the prevalence of patients eligible for treatment with PCSK9 inhibitors in a real-life clinical setting in Italy before and after the recent enlargement of reimbursement criteria. Secondly, we aimed to estimate (i) the number of patients who, based on Italian national rules, currently do not fulfill the prescription criteria for PCSK9 inhibitors and (ii) their residual distance to the recommended LDL-C target, (iii) the number of patients whose LDL-C levels could be optimized by low-cost tailored therapeutic interventions for individual patients, and (iv) the number of high CV risk patients who are not allowed to access to treatment with PCSK9 inhibitors based on the national rules but who could reach their own LDL-C target with PCSK9 inhibitors.

## 2. Methods

### 2.1. Study Design and Participants

For the study, we considered the medical records of the patients (N. 6241) admitted at the Lipid Clinic of the University Hospital of Bologna (Italy) from December 2017 to December 2021.

Patients were first classified based on their diagnosis (either primary or secondary dyslipidemia, and prevalent plasma lipid phenotype) and considering their estimated CV risk, as recommended by the most recent guidelines of the European Atherosclerosis Society (EAS) and the European Cardiology Society (ESC) [9]. They were further stratified according to whether they were allowed or not allowed access to treatment with PCSK9 inhibitors based on national previous and current (from June 2022) reimbursement rules issued by the Italian Medicines Agency (AIFA).

Till June 2022, patients eligible for prescription of PCSK9 inhibitors were (i) individuals aged <80 years in primary prevention for CVD with a definitive diagnosis of HeFH (based either on a positive genetic test result or a Dutch Lipid Clinical Network Score (DLCNS) ≥ 8) and LDL-C > 130 mg/dL despite maximally tolerated LLT; (ii) patients in secondary prevention for CVD and LDL-C > 100 mg/dL despite maximally tolerated LLT; (iii) patients with LDL-C > 100 mg/dL despite maximally tolerated LLT and type 2 diabetes with either at least one other CV risk factor or renal impairment and/or signs of retinopathy; (iv) patients with homozygous FH (HoFH) and LDL-C > 100 mg/dL despite maximally tolerated LLT. Patients who did not fulfill the criteria for the reimbursement of PCSK9 inhibitors were (i) non-FH individuals on maximally tolerated LLT and without any established CVD but who were estimated to be at high CV risk; (ii) FH individuals without any established CVD and LDL-C between 130 mg/dL and 70 mg/dL despite maximally tolerated LLT; (iii) individuals with CVD and LDL-C between 100 mg/dL and 55 mg/dL despite maximally tolerated LLT [10,11].

From June 2022, patients in secondary prevention for CVD and LDL-C between 100 mg/dL and 70 mg/dL despite maximally tolerated LLT, can be treated with fully reimbursed PCSK9 inhibitors, as well [12].

The distance to the target level for LDL-C was calculated as the difference between the actual value of LDL-C and the target level for LDL-C as recommended by the latest ESC/EAS guidelines [9].

The difference between AIFA reimbursement criteria and the EMA (based on the ESC/EAS guidelines) and FDA (based on the AHA/ACC guidelines [13]) recommendations for PCKS9 inhibitors prescription is resumed in Table 1.

Patients eligible for treatment with PCSK9 inhibitors but who did not fulfill the reimbursement criteria were treated with alternative therapeutic interventions, including (i) the intensification of therapeutic lifestyle interventions, paying particular attention to the optimization of body weight and increasing aerobic physical activity; (ii) switching to a more efficacious statin and/or increasing statin dose (where possible and tolerable); (iii) the prescription of fenofibrate (especially in patients with concomitant high TG levels); (iv) the prescription of dietary supplements other than red yeast rice (i.e., not acting as inhibitors of the 3-hydroxy-3-methyl-Coenzyme A reductase).

In patients with statin intolerance, more intensive and personalized counseling (increased number of visits and virtual contacts) and other pragmatic therapeutic strategies were also implemented [14]: (i) the use of alternate-day low-dose of statins (e.g., rosuvastatin 5 mg every other day); (ii) the use of lipid-lowering nutraceuticals (mainly low-dose red yeast rice in association with berberine or phytosterols, or bergamot and artichoke standardized extracts); (iii) the use of fenofibrate (mainly in patients with concomitant high TG levels), alone or associated with statins; (iv) strategies to improve statin tolerability (i.e., switch to a less powerful statin, shift statin from the evening to the morning in patients with nocturnal cramps, supplementation with high-dose coenzyme Q10 or magnesium salts).

Considering the 6-month reduction in LDL-C observed in our sample, we provided a simulation of the number of individuals whose serum lipids could be enhanced (according to the ESC/EAS guidelines) if AIFA reimbursement criteria were expanded.

The study protocol has been approved by the Institutional Ethical Board of the University Hospital of Bologna (Code: LLD-RP2018) and performed in accordance with the ethical standards laid down in the 1964 Declaration of Helsinki and its later amendments. All involved individuals signed an informed consent form prior to their inclusion in the study.

### 2.2. Assessments

#### 2.2.1. Clinical Data and Anthropometric Measurements

Information gathered in the patients’ history included presence of CVD and other systemic diseases, allergies, and medications.

Height and weight were respectively measured to the nearest 0.1 cm and 0.1 kg, with subjects standing erect with their eyes directed straight while wearing light clothes and having bare feet. BMI was calculated as body weight in kilograms, divided by height squared in meters (kg/m^2^).

#### 2.2.2. Laboratory Analyses

Biochemical analyses were carried out on venous blood withdrawn after overnight fasting (at least 12 h). The serum was obtained by the addition of disodium ethylenediaminetetraacetate (Na_2_EDTA) (1 mg/mL) and blood centrifugation at 3000 RPM for 15 min at 25 °C.

Immediately after centrifugation, trained personnel performed laboratory analyses according to standardized methods [15]. The following parameters were directly assessed: total cholesterol (TC), triglycerides (TG), high-density lipoprotein cholesterol (HDL-C), apolipoprotein B-100 (Apo B-100), fasting plasma glucose (FPG), creatinine, creatine phosphokinase (CPK), gamma-glutamyl transferase (GGT), alanine transaminase (ALT) and aspartate transaminase (AST).

LDL-C was obtained by the Friedewald formula. When plasma TG levels were higher than 400 mg/dL, the Sampson formula was used [16]. The glomerular filtration rate (eGFR) was estimated by the Chronic Kidney Disease Epidemiology Collaboration (CKD-epi) equation [17].

#### 2.2.3. Statin Intolerance

Statin intolerance was defined as presence of statin-associated muscle symptoms (i.e., increase in serum creatine phosphokinase (CPK) by more than 5-time the upper limit of normal (ULN), myalgia, rhabdomyolysis) with at least two different statins (one at the minimum effective dose) [18].

#### 2.2.4. Assessment of Safety and Tolerability

Safety and tolerability were evaluated through continuous monitoring during the study, in order to detect any adverse event, clinical safety, laboratory findings, vital sign measurements, and physical examinations. An independent expert clinical event committee was appointed in order to categorize the adverse events that could possibly be experienced during the study as not related, unlikely related, possibly related, probably related, or definitely related to the tested treatment.

### 2.3. Statistical Analysis

Data were encoded and statistically analyzed by the use of the Statistical Package for the Social Sciences (SPSS) 27.0, version for Windows (IBM Corporation, Armonk, NY, USA). Categorical variables were reported as absolute numbers and percentages. Continuous variables were reported as mean ± standard deviation (SD) and compared by Analysis of Variance (ANOVA) followed by Levene’s homogeneity of variance test and Student’s *t*-test after data normalization (when needed). The target achievement percentage of LDL-C was finally calculated. A *p* level < 0.05 was considered significant for all tests.

## 3. Results

The flowchart of the study has been reported in Figure 1.

For the purpose of this analysis, from an initial sample of 6231 individuals (Men: 3027; Women: 3204), we firstly excluded 3891 patients with estimated low-to-moderate CV risk. Patients with pure hypertriglyceridemia (N. 359), isolated elevated lipoprotein(a) levels (N. 293), or dyslipidaemias due to secondary causes (e.g., extreme dietary habits, hypothyroidism, uncontrolled type 2 diabetes, nephrotic syndrome, multiple myeloma, iatrogenic causes; N. 273) were also excluded from the analysis. Finally, according to the pre-specified study’s design, we excluded patients with HeFH or with high-to-very high CV risk of reaching LDL-C goal with first-level LLT (N. 439).

The remaining individuals (N. 986; Men: 475, Women: 511) were potential candidates for treatment with PCSK9 inhibitors. Based on national reimbursement rules issued by AIFA, 180 patients (Men: 88; Women: 92) were allowed access to treatment with PCSK9 inhibitors while 806 patients were not allowed. The reasons why these patients did not fulfill the prescription criteria for PCSK9 inhibitors were that they were non-He-FH hypercholesterolemic individuals with high risk of developing CVD (N. 316), HeFH individuals free from CVD with LDL-C between 130 mg/dL and 70 mg/dL (N. 178) or patients with very high CV risk with LDL-C between 100 mg/dL and 55 mg/dL (N. 312).

In a relevant percentage of cases, in patients who did not fulfill the prescription criteria for PCSK9 inhibitors the alternative therapeutic interventions provided the necessary reduction in LDL-C to achieve the LDL-C target level (Table 2). No patient experienced any subjective or laboratory adverse event.

Obviously, patients treated with PCSK9 inhibitors also experienced significant improvements in serum levels after 6 months of treatment (Table 3).

Overall, 98% of statin-tolerant and 74% of statin-intolerant patients achieved the target level for LDL-C as recommended by the ESC/EAS guidelines. Both sub-groups had an improvement in LDL-C of at least 50% and LDL-C target was reached respectively by 89% and 85% of statin-tolerant and statin-intolerant patients. Moreover, in both sub-groups total cholesterol, LDL-C, very LDL-C, apolipoprotein B, and lipoprotein(a) significantly improved. No side effects were registered during the first 6 months of treatment.

When considering the mean change in LDL-C experienced by our patients during treatment with PCSK9 inhibitors, it is likely that, if all the pre-specified patients with high CV risk had been on treatment, the LDL-C target would have been respectively reached by 95% of statin-tolerant and 92% of statin-intolerant individuals.

In our cohort, the current rules for PCSK9 inhibitor reimbursement increase the number of patients who could benefit from this treatment from 180/976 to 322/976 (+14.4%), reaching 32.7% of the eligible patients.

A hypothetical further expanded access to treatment with PCSK9 inhibitors for (i) patients with probable HeFH and LDL-C > 100 mg/dL despite high-intensity LLT (i.e., maximal tolerated dose of high-efficacy statins and ezetimibe) and (ii) Non-HeFH hypercholesterolemic patients with high risk of CVD and LDL-C > 130 mg/dL despite maximally tolerated LLT would bring the number of patients who may benefit from treatment with PCSK9 inhibitors from 322/986 (32.7%) to 472/986 (47.9%). However, in our experience, the intensification of LLT with repeated and personalized patient counseling added to any currently available alternative therapeutic interventions to PCSK9 inhibitors could reduce the prevalence to 381/986 (38.6%).

## 4. Discussion

CVD is still the leading cause of disability and death in developed countries and LLT is one of the milestones in CV risk reduction [19,20]. Despite the efficacy of the first-line LLT, namely statins and ezetimibe, a relatively large number of high CV risk individuals maintain an LDL-C level far from goal [21,22].

In this context, PCSK9 inhibitors have been shown to substantially reduce both LDL-C level (−51% [95% confidence interval (CI): −61%, −41%]) and the overall relative risk (RR) of major CV events [RR = 0.80, 95%CI: 0.73, 0.87)] [23] with high safety profile, even in frail patients [24,25].

In the real-life setting we explored, we identified a large number of individuals potentially needing treatment with PCSK9 inhibitors but who did not fulfill the prescription criteria for PCSK9 inhibitors issued by AIFA before June 2022. The intensification of therapeutic interventions alternative to PCSK9 inhibitors made it possible to reach a further LDL-C reduction of at least 50% with respect to baseline in more than 60% of statin-tolerant patients and in around 40% of statin-intolerant patients. However, the LDL-C target was reached only by around 40% of statin-tolerant patients and by less than 30% of statin-intolerant patients. According to our observations, low-cost lipid-lowering therapeutic interventions can nearly half the number of individuals potentially needing treatment with PCSK9 inhibitors among the patients who are not allowed access to the treatment. However, the main limitation of this approach is that it strongly depends on the experience of the physician. The updated AIFA rules for PCSK9 inhibitors reimbursement increase the number of patients who could benefit from this treatment by 14.4%, including more patients in secondary prevention for cardiovascular diseases. Certainly, the availability of new oral lipid-lowering drugs (e.g., bempedoic acid) could further reduce the gulf between the number of patients needing to reach lower LDL-C levels and the number of patients who reached them [26]. In particular, bempedoic acid is a first-in-class prodrug activated by a liver enzyme not present in skeletal muscle and inhibiting the ATP-citrate lyase, an enzyme upstream of β-hydroxy β-methylglutaryl-coenzyme A reductase in the cholesterol biosynthesis pathway: we could expect by its use a further LDL-C reduction of about 20% [27]. This finding is relevant since it could improve the cost-benefit ratio of PCSK9 inhibitors in case the access to treatment was expanded to additional patient categories. In agreement with available data, after 6-month treatment with PCSK9 inhibitors, we also observed a significant improvement in lipid fractions other than LDL-C, with an optimal safety profile even in statin-intolerant patients [28].

Certainly, PCSK9 inhibitors are relatively expensive and the cost-benefit ratio has to be carefully evaluated and precise ethical standards must be defined [29]. We estimated that extended criteria that are more adherent to the international guidelines’ recommendations could nearly triple the prescription of PCSK9 inhibitors, and the costs would increase as a consequence. Note that in Italy, the cost of monthly therapy with PCSK9 inhibitors is quite contained when compared with other high-income countries. Moreover, the cost-effectiveness ratio of the treatment with PCSK9 inhibitors is favorable when patients to be treated are individuals with very high CV risk who are far from the LDL-C target despite the optimization of background LLT [30,31]. However, based on our observations, waiting for bempedoic acid will become reimbursable, and careful use of the available lipid-lowering treatments could significantly reduce the number of subjects to be treated with PCSK9 inhibitors. In the next months, probably, inclisiran a novel posttranscriptional gene silencing therapy that inhibits PCSK9 synthesis by RNA interference will be also available in Italy. The main advantage of this drug is the possibility to administer it each 6 months. A cost-effectiveness comparison with inclisiran is currently not easy since the final reimbursement price of this drug in Italy has not yet been established, while inclisiran could be mildly less effective than PCSK9 inhibitors in terms of LDL-C lowering and its cardiovascular disease preventive effect has not yet been demonstrated [32].

Of course, our study has some relevant limitations. Firstly, according to the study’s design, the standardization of the therapeutic intervention alternative to treatment with PCSK9 inhibitors is not possible without the concept of tailored medicine for individual patients fails. Moreover, our observations were based on the last available patients’ laboratory exams. However, some patients could have further improved or worsened their blood lipid control after the last visit. Then, our lipid clinic is a hub for the management of HeFH and other severe hyperlipidaemias, so our cohort could be different compared to one of the cardiology units, managing more coronary heart disease patients, for whom PCSK9 inhibitors are more easily reimbursed. On the other side, this is an example of a real practice scenario. Finally, rigorous cost-effectiveness analyses have been not carried out, because of the heterogeneity of the enrolled patients and because the PCSK9 cost is progressively reduced since their first authorization in Italy and the formal price is further mandatorily reduced at the moment of the hospital purchase. However, we can currently state that the ex-factory cost of evolocumab and alirocumab in Italy is 21,734 Euros (+VAT) per pen (i.e., 43,468 Euro per month). The final price paid by the hospital could be reduced even by a further 15%. Considering that the mean number needed to treat (NNT) to avoid a major cardiovascular outcome with PCSK9 inhibitors is meanly 28 on 3 years of follow-up [33], we could conclude that the global added cost to prevent one major event is about 124,144 Euro. The NNT seems to be dramatically decreased with longer follow-ups, reaching the value of 5 after 5 years [33]. As a consequence, the added cost for one saved life every 5 years is about 88,675 Euros. Of course, these costs are still very high, but this makes the PCSK9 cost one of the lowest in the world [30,31]. Comparisons with other settings are hard, particularly when considering the different treatment approaches of European and US guidelines in terms of LDL-C reduction to be reached [34].

## 5. Conclusions

Based on our observations, low-cost tailored therapeutic interventions for individual patients can significantly reduce the number of patients potentially needing treatment with PCSK9 inhibitors among those who are not allowed access to the treatment.

The recently enlarged reimbursement criteria for PCSK9 inhibitors could mildly improve the possibility of adequately managing high-risk hypercholesterolemic subjects in the setting of an outpatient lipid clinic.

However, the application of strict reimbursement criteria for PCSK9 inhibitors still limits the rate of high CV risk patients able to reach in real life the LDL-C goal recommended by the international guidelines. The cost-effectiveness of this consequence has to be evaluated in large longitudinal pharmacoeconomic studies.

## Figures and Tables

**Figure 1 jcm-11-04701-f001:**
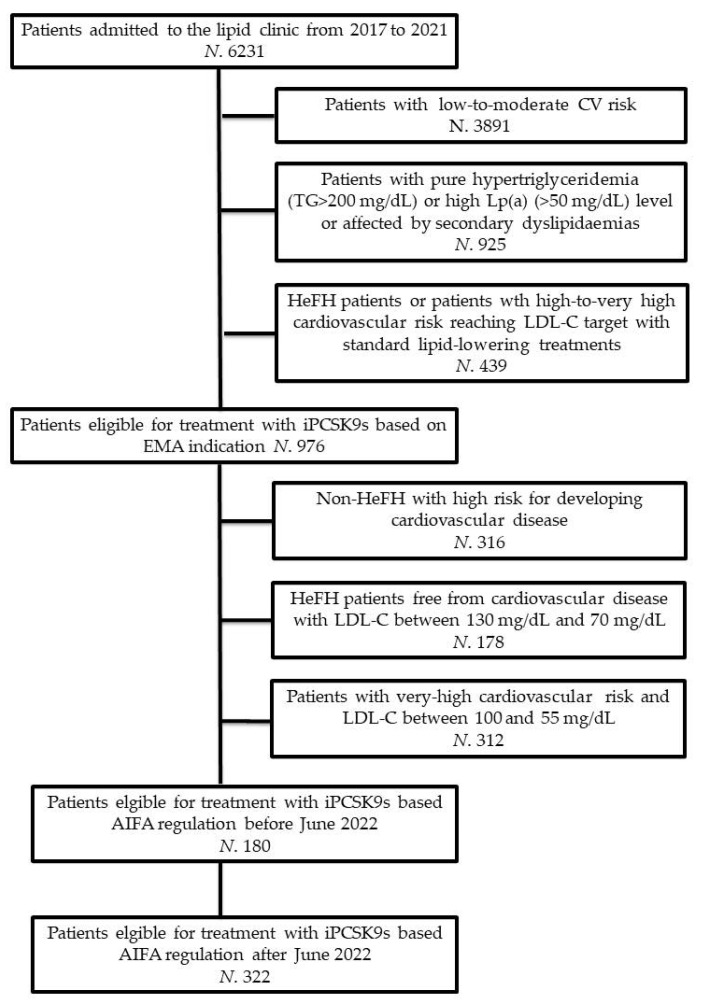
Flowchart of the study. AIFA = Italian Medicines Agency; EMA = European Medicines Agency; HeFH = Heterozygous familial hypercholesterolemia; iPCSK9s = Proprotein convertase subtilisin/kexin type 9 inhibitors; LDL-C = Low-density lipoprotein cholesterol; Lp(a) = Lipoprotein(a).

**Table 1 jcm-11-04701-t001:** AIFA reimbursement rules and EMA and FDA prescription criteria for the PCSK9 inhibitors.

	AIFA (Old)	AIFA (Current)	EMA (ESC/EAS)	FDA (AHA/ACC)
Age	≤80 years old	≤80 years old	No limit beyond patient fitness	No limit beyond patient fitness
Primary prevention	Patients with definitive diagnosis of HeFH (genetic test result or DLCNS ≥ 8) and LDL-C > 130 mg/dL despite maximally tolerated LLT	Patients with definitive diagnosis of HeFH (genetic test result or DLCNS ≥ 8) and LDL-C > 130 mg/dL despite maximally tolerated LLT	Patients with FH and another major cardiovascular risk factor (very-high risk FH patients) with LDL-C > 55 mg/dL despite maximally tolerated LLT Maybe considered in non-FH very high-risk patients with LDL-C > 55 mg/dL despite maximally tolerated LLT	Patients with severe primary hypercholesterolemia (LDL-C level ≥ 190 mg/dL) with LDL-C ≥ 100 mg/dL despite maximally tolerated LLT
LDL-C target	Not reported	Not reported	LDL-C reduction ≥50% from baseline and LDL-C < 55 mg/dL	LDL-C reduction ≥50% from baseline (No treat-to-target approach)
Secundary prevention	Patients with LDL-C > 100 mg/dL despite maximally tolerated LLT	Patients with LDL-C > 70 mg/dL despite maximally tolerated LLT	Patients with atherosclerosis-related cardiovascular disease and with LDL-C > 55 mg/dL despite maximally tolerated LLT	Patients with multiple major atherosclerosis-related cardiovascular events or 1 major ASCVD atherosclerosis-related cardiovascular event and multiple high-risk conditions with LDL-C ≥ 70 mg/dL despite maximally tolerated LLT
LDL-C target	Not reported	Not reported	LDL-C reduction ≥50% from baseline and LDL-C < 55 mg/dL If a second vascular events occur before 2 years, then consider LDL-C < 40 mg/dL	LDL-C reduction ≥50% from baseline (No treat-to-target approach)
Type 2 Diabetes	Patients with LDL-C > 100 mg/dL despite maximally tolerated LLT and type 2 diabetes with either at least one other CV risk factor or renal impairment and/or signs of retinopathy	Patients with LDL-C > 100 mg/dL despite maximally tolerated LLT and type 2 diabetes with either at least one other CV risk factor or renal impairment and/or signs of retinopathy	Patients at high-risk and with LDL-C > 70 mg/dL despite maximally tolerated LLT. Patients with type 2 diabetes at very high-risk and with LDL-C > 55 mg/dL despite maximally tolerated LLT.	Not considered if not included in the above listed categories
LDL-C target	Not reported	Not reported	LDL-C reduction <70 mg/dL If target organ damage, early-onset type 1 diabetes (>20 years), or more than 2 other cardiovascular risk factors, then consider <55 mg/dL	LDL-C reduction ≥50% from baseline if high-risk patients (No treat-to-target approach)

AIFA = Agenzia Italiana del Farmaco (Italian Drug Agency), DLCNS = Dutch Lipid Clinical Network Score, EMA = European Medicines Agency, FDA = Food and Drug Administration, LLT = Lipid-lowering treatment.

**Table 2 jcm-11-04701-t002:** Therapeutic achievement after 6 months of alternative interventions to PCSK9 inhibitors, in patients who were not allowed access to treatment with PCSK9 inhibitors based on national reimbursement rules issued by AIFA before June 2022.

		LDL-C (mg/dL)		Portion of Patients Who Achieved a Reduction in LDL-C > 50% (%)	Portion of Patients Who Achieved the LDL-C Target Level (%)
		At Baseline; with Maximally Tolerated LLT	Therapeutic Efficacy of Alternative Interventions to PCSK9 Inhibitors		
Non-He-FH hypercholesterolemic individuals with high risk of developing CVD	Patients undergoing statin treatment	118 ± 11	90 ± 9 *	64	41
	Statin-intolerant patients	141 ± 12	122 ± 10 *	42	29
HeFH individuals free from CVD with LDL-C between 130 mg/dL and 70 mg/dL	Patients undergoing statin treatment	97 ± 7	89 ± 5 *	62	44
	Statin-intolerant patients	111 ± 10	92 ± 8 *	39	28
Patients with very high CV risk with LDL-C between 100 mg/dL and 55 mg/dL	Patients undergoing statin treatment	69 ± 5	59 ± 5 *	68	49
	Statin-intolerant patients	84 ± 6	71 ± 4 *	43	27

* *p* < 0.05 versus baseline. CV = Cardiovascular; CVD = Cardiovascular disease; HeFH = Heterozygous familial hypercholesterolemia; LDL-C = Low-density lipoprotein cholesterol; LLT = Lipid—lowering therapy; PCKS9 = Proprotein convertase subtilisin/kexin type 9.

**Table 3 jcm-11-04701-t003:** Effect of 6-month treatment with PCKS9 inhibitors in statin-tolerant and statin-intolerant patients.

	Statin-Tolerant Patients (N. 128)		Statin-Intolerant Patients (N. 52)	
	Pre-Treatment	Post-Treatment	Pre-Treatment	Post-Treatment
Age (years)	66 ± 8		62 ± 11	
Body mass index (kg/m^2^)	26.6 ± 3.8	27.1 ± 3.9	27.6 ± 3.6	26.7 ± 3.2 *
Fasting plasma glucose (mg/dL)	99 ± 14	99 ± 18	102 ± 22	101 ± 19
Serum uric acid (mg/dL)	5.7 ± 1.3	5.6 ± 1.3	5.1 ± 1.2	5.9 ± 1.1 *
eGFR (CKD-EPI, mL/min/1.73 m^2^)	76.2 ± 21.1	75.3 ± 21.1	83 ± 15	85 ± 13
Total cholesterol (mg/dL)	201 ± 49	131 ± 45 *	194 ± 48	125 ± 39 *
LDL-cholesterol (mg/dL)	149 ± 47	52 ± 18 *	143 ± 32	47 ± 12 *
HDL-cholesterol (mg/dL)	51 ± 10	53 ± 11	53 ± 12	55 ± 12
Triglycerides (mg/dL)	162 ± 52	150 ± 42	143 ± 83	131 ± 31
VLDL-cholesterol (mg/dL)	32 ± 20	26 ± 16 *	28 ± 17	21 ± 10 *
Apolipoprotein B (mg/dL)	121 ± 41	59 ± 24 *	101 ± 28	55 ± 15 *
Lipoprotein(a) (mg/dL)	64 ± 21	56 ± 22 *	77 ± 29	65 ± 23 *
AST (mg/dL)	27 ± 11	27 ± 14	26 ± 6	29 ± 9
ALT (mg/dL)	27 ± 17	26 ± 14	26 ± 10	30 ± 12
gamma-GT (mg/dL)	37 ± 17	35 ± 20	33 ± 19	34 ± 15
CPK (mg/dL)	203 ± 57	253 ± 61	195 ± 59	196 ± 78

* *p* < 0.05 versus pre-PCSK9 inhibitors administration. ALT = Alanine transaminase; AST = Aspartate transaminase; CKD-EPI = Chronic Kidney Disease Epidemiology Collaboration; CPK = Creatinine phosphokinase; eGFR = Estimated glomerular filtration rate; gamma-GT = gamma-glutamyl transferase; HDL = High-density lipoprotein; LDL = Low-density lipoprotein; VLDL = Very-low density lipoprotein.

## Data Availability

Data supporting the findings of this analysis are available from the University of Bologna. Data are available from the authors with the permission of the University of Bologna.

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
