# Peer review of "Management of High-Risk Hypercholesterolemic Patients and PCSK9 Inhibitors Reimbursement Policies: Data from a Cohort of Italian Hypercholesterolemic Outpatients"

_jcm, 2022, doi:10.3390/jcm11164701_

Round 1

Reviewer 1 Report

Title: no comments

Summary: no comments.

Introduction: with the intention of making the article clearer, especially the interpretation of the results for non-Italian readers I suggest:

a) refer to the original studies (references 3 and 4);

b) for the EMA criteria refer to what are the LDL cholesterol targets derived from the ESC/EAS Guide 2019 to indicate an mab-PCSK9;

c) for the FDA criteria refer to what are the thresholds derived from the AHA/ACC 2018 Guide to indicate a mab-PCSK9;

c) mention in more detail why the European approach is different from the American one (therapeutic targets vs therapeutic thresholds);

(d) finally, to specify more clearly the AIFA criteria before and after enlargement. A table might be convenient for comparing the criteria of the EMA, FDA, and AIFA.

Methods: a) I suggest specifying how LDL cholesterol was calculated in individuals with hypertriglyceridemia; b) I suggest adding reference to the definition of statin intolerance.

Results: no observations

Discussion: a) I suggest adding at least what is the current price of mab-PCSK9 in Italy and a at least approximate cost-effectiveness analysis (at least the cost of a year of life with quality at the current price of mabs-PCSK9) in Italy.

Author Response

Title: no comments

Summary: no comments.

Introduction: with the intention of making the article clearer, especially the interpretation of the results for non-Italian readers I suggest:

  1. a) refer to the original studies (references 3 and 4);
  2. b) for the EMA criteria refer to what are the LDL cholesterol targets derived from the ESC/EAS Guide 2019 to indicate an mab-PCSK9;
  3. c) for the FDA criteria refer to what are the thresholds derived from the AHA/ACC 2018 Guide to indicate a mab-PCSK9;
  4. c) mention in more detail why the European approach is different from the American one (therapeutic targets vs therapeutic thresholds);

(d) finally, to specify more clearly the AIFA criteria before and after enlargement. A table might be convenient for comparing the criteria of the EMA, FDA, and AIFA.

A: We have included original paper as per ref 3 and 4. Then, we have tried to resume all other requests in a single table.

Methods: a) I suggest specifying how LDL cholesterol was calculated in individuals with hypertriglyceridemia; b) I suggest adding reference to the definition of statin intolerance.

A: We are grateful to the reviewer for these suggestion. We have now explained how LDL-C was calculated in individuals with hypertriglyceridemia and replaced the reference related to the statin intolerance definition

Results: no observations

Discussion: a) I suggest adding at least what is the current price of mab-PCSK9 in Italy and a at least approximate cost-effectiveness analysis (at least the cost of a year of life with quality at the current price of mabs-PCSK9) in Italy.

A: We thanks the reviewer for his valuable suggestion. We have now added some raw cost-effectiveness analyses, based on the available data, with the limitation reported in the paper.

Reviewer 2 Report

I read with interest the article titled "Management of high-risk hypercholesterolemic patients and PCSK9 inhibitors of reimbursement policies: data from a cohort of Italians hypercholesterolemic outpatient".

This is a quality and potentially interesting article. The results are presented correctly, and the discussion is meaningful with the limits of the study correctly stated.

I advise in the introduction and/or in the discussion to mention newer treatment options for different CV patients with unregulated dyslipidemia with inclisiran and ev. advantages / limitations inclisiran in relation to PCSK9 inhibitors.

My main suggestion to the authors is to list the latest "non-statin" treatment options for dyslipidemia with inclisiran. Also, I advise that they state the possible pharmacoeconomic advantages or disadvantages of treatment with PCSK9 inhibitors vs. inclisiran (number of drug applications, price...)

Author Response

I read with interest the article titled "Management of high-risk hypercholesterolemic patients and PCSK9 inhibitors of reimbursement policies: data from a cohort of Italians hypercholesterolemic outpatient".

This is a quality and potentially interesting article. The results are presented correctly, and the discussion is meaningful with the limits of the study correctly stated.

A: We are grateful to the reviewer for the positive comments on our paper.

I advise in the introduction and/or in the discussion to mention newer treatment options for different CV patients with unregulated dyslipidemia with inclisiran and ev. advantages / limitations inclisiran in relation to PCSK9 inhibitors.

My main suggestion to the authors is to list the latest "non-statin" treatment options for dyslipidemia with inclisiran. Also, I advise that they state the possible pharmacoeconomic advantages or disadvantages of treatment with PCSK9 inhibitors vs. inclisiran (number of drug applications, price...)

A: As per reviewer suggestion, we have included some comments on the comparison between new non-statin treatment options for dyslipidaemia (including inclisiran) and their potential role compared to PCKS9 inhibitors. However, both bemepedoic acid and incisiran are not yet reimbursed in Italy, so that we can only suppose their potential role in lipid disorders management and cost-effectiveness. All of this has been now more clearly discussed in the discussion section.

Round 2

Reviewer 1 Report

Paper could be published; suggestions have been made according with my suggestions.

Thank you.